# Encoding Text Information with Graph Convolutional Networks for Personality Recognition

**Zhe Wang, Chun-Hua Wu *, Qing-Biao Li, Bo Yan and Kang-Feng Zheng**

School of Cyberspace Security, Beijing University of Posts and Telecommunications, Beijing 100876, China; wangxiaozhe@bupt.edu.cn (Z.W.); liqingbiao@bupt.edu.cn (Q.-B.L.); mjkbyb@bupt.edu.cn (B.Y.); kfzheng@bupt.edu.cn (K.-F.Z.)

* Correspondence: wuchunhua@bupt.edu.cn; Tel.: +86-186-0010-5255

**Abstract:** Personality recognition is a classic and important problem in social engineering. Due to the small number and particularity of personality recognition databases, only limited research has explored convolutional neural networks for this task. In this paper, we explore the use of graph convolutional network techniques for inferring a user's personality traits from their Facebook status updates or essay information. Since the basic five personality traits (such as openness) and their aspects (such as status information) are related to a wide range of text features, this work takes the Big Five personality model as the core of the study. We construct a single user personality graph for the corpus based on user-document relations, document-word relations, and word co-occurrence and then learn the personality graph convolutional networks (personality GCN) for the user. The parameters or the inputs of our personality GCN are initialized with a one-hot representation for users, words and documents; then, under the supervision of users and documents with known class labels, it jointly learns the embeddings for users, words, and documents. We used feature information sharing to incorporate the correlation between the five personality traits into personality recognition to perfect the personality GCN. Our experimental results on two public and authoritative benchmark datasets show that the general personality GCN without any external word embeddings or knowledge is superior to the state-of-the-art methods for personality recognition. The personality GCN method is efficient on small datasets, and the average F1-score and accuracy of personality recognition are improved by up to approximately 3.6% and 2.4–2.57%, respectively.

**Keywords:** personality recognition; word co-occurrence; information sharing; correlation; personality GCN

## 1. Introduction

Personality is the combination of a person's emotion, behavior, and motivation and the characteristics of a person's thought patterns. The ability to detect one's personality traits automatically has important research significance and value [1]. Our personality has a major impact on our lives, affecting our life choices, health, and well-being, along with our desires and preferences [2]. In marketing, it can be useful in recommending books, music, etc., to users based on their personality. In the field of psychology, it may be applied to social network data to understand user behavior. In recruiting, it can also help job recruiters select appropriate employees [3–5]. Additionally, personality, as a stable set of psychological characteristics, has been widely used in the field of security research to prevent social engineering attacks. In the field of security, personality is used to discover the principles of how attackers recognize and use personality. Blocking or interfering with personality recognition is an important prerequisite for defending against social engineering attacks.

Personality theory is divided into six schools of psychoanalysis, traits, biology, humanism, behaviorism, and cognition [6,7]. The most commonly used personality model is the Big Five personality traits model, which describes personality in five aspects: extroversion, neuroticism, agreeableness, conscientiousness, and openness [8], as detailed in Table 1. The Big Five personality model is a well-experimented and well-scrutinized standard of personality structure used by researchers in recent years [9,10]. Therefore, the majority of the studies use the Big Five personality measure for classification. Personality is a stable psychological characteristic. The user's behavior state will vary with the time and environment, but the user's personality is stable. According to the different behaviors of users, we extract feature information to identify the user's personality. Traditionally, personality recognition depends on the user's profile information, status updates, texts, and so on [11–14]. Most of this information comes from textual expression. Analyzing the text data sent by the user is the most effective and accurate way to obtain the user's personality. Therefore, the analysis of user text data allows us to determine important personality traits.

**Table 1.** Description of the Big Five personality traits.

| Traits | Description |
| --- | --- |
| Extroversion (EXT) | Is this person outgoing, talkative, and energetic or reserved and solitary? |
| Neuroticism (NEU) | Is this person sensitive and nervous or secure and confident? |
| Agreeableness (AGR) | Is this person trustworthy, outspoken, generous, and humble or unreliable, complex, meager, and boastful? |
| Conscientiousness (CON) | Is this person efficient and organized or careless and sloppy ? |
| Openness (OPN) | Is this person creative and curious or dogmatic and cautious? |

In recent years, there have been a number of studies that have applied machine learning and convolutional neural networks to personality recognition. Personality recognition from social media [15], especially Facebook sentiment analysis, has been widely welcomed. However, most studies on personality recognition are focused on feature extraction based on data collection and preprocessing [16–18]. Hence, research can focus on creating better models and architectures, not just on data acquisition and preprocessing. The classification algorithms of traditional machine learning and multilayer neural networks are mostly used to determine personality categories. However, the accuracy and F1-score of automatically classifying personality traits by analyzing user social information are currently low. The most important consideration is that although most researchers consider these five personality traits to be independent of each other, several studies [19,20] claim that there is a certain correlation between these traits, and it is inaccurate to build five completely independent classifiers. Most studies ignore the important question of the correlation between the five personality traits.

Consequently, for the above problems, this paper uses graph convolutional networks (GCN) to capture users' text information. It is a recently proposed neural architecture for representing relational information. We use the feature information sharing method to add correlations between the five personality traits in the model. We show that personality graph convolutional networks (personality GCN) can effectively use text information to detect user personality and can significantly improve performance, even on small datasets.

The main contributions of this paper are summarized below:

- We propose a novel personality recognition method based on the graph convolutional neural network method. According to our understanding, this is the first personality recognition study to model the entire user text information corpus as a heterogeneous graph and learn user, word, and document embeddings together through graph neural networks.

- This paper focuses on the correlation between the five personality traits. Unlike previous research, the personality GCN model shares a set of basic feature information during user personality recognition, rather than using five sets of features for classification. The feature information sharing fully considers the correlation between the five personality characteristics during training. We propose an innovation model that is completely different from previous research.
- The results on two public and authoritative benchmark datasets show that our method's accuracy and F1-score outperform the latest personality recognition methods, without using pretrained word embeddings or external knowledge.

The rest of this paper is organized as follows. Section 2 presents the work carried out in the field of personality recognition. Section 3 introduces the background of the GCN. Section 4 describes a graph convolution model for personality recognition. Section 5 presents experiments that analyze the effectiveness of our model. Section 6 concludes and presents future work.

## 2. Related Work

In previous studies, the most widely used personality representation method was self-reporting, which meant that data were collected from users who filled out standardized questionnaires [17]. Therefore, this method is affected by user subjectivity. At present, the method of automatic personality recognition based on big data analysis has become the main research point. Social networks are the main source of people's opinions and thoughts, and analyzing them can provide valuable insights for personality recognition.

Traditionally, the identification of users' personality based on social information has been studied as a text classification problem [21–24], focusing on personality traits or textual indicators that indicate the personality of the user. Generally, features are extracted from text, like linguistic inquiry and word count (LIWC) [25], Mairesse [26], and the Medical Research Council (MRC) [27], which are then fed into standard machine learning classifiers such as decision tree (DT), support vector machine (SVM), and naive Bayes. On the other hand, learning word embeddings and representing them as vectors (using GloVe or word2vec) is also a common method [28].

The work in [29] introduced personality recognition tasks from multiple computational perspectives by comparing the use of written papers and speech corpora as input data. Contrary to the use of psycholinguistic-driven features in [29] and other literature, the work in [30] used the naive Bayesian model and SVM model to classify personality traits using n-gram models. Accuracy assessment based on individual blog collections could reach up to 65%. The work in [31] used charand POSn-gram models, and the work in [29] used TF-IDF counting and style features. However, to our knowledge, there is little ongoing research based on the correlation between the five personality traits in texts. In other words, the research on text personality analysis of social network users is inaccurate when the correlation between the five personality traits of users is ignored.

Recently, a new research direction called graph neural network or graph embeddings has attracted widespread attention [32]. Graph neural networks have effectively completed tasks with rich relational structures and can retain the global structure information of graphs in graph embeddings [33]. We used a recently proposed graph embedding framework, graph convolutional networks (GCN) [34], to capture these relationships. Therefore, this paper proposes a personality GCN representation model for user personality recognition.

## 3. Background: Graph Convolutional Networks

In this section, this paper gives a brief overview of GCN. Extracting the spatial features of the topological graph is the essential purpose of GCN, and the theoretical basis of GCN is the spectral domain. The idea is to use graph theory to implement convolution operations on topological graphs [35]. Throughout the research process, first, scholars who studied graph signal processing (GSP) defined Fourier transformation on graphs, then defined convolution on graphs, and finally, proposed a graph convolutional network in combination with deep learning.

Graph convolutional networks are multi-layer neural networks that can be operated directly on the graph and derive the node's embeddings vector based on the properties of its neighborhood [36]. Usually, consider a graph $G = (V, E)$, where $V(|V| = n)$ and $E$ are sets of nodes and edges, respectively. Assume that each node is connected to itself, that is any $v(v, v) \in E$. Let $X \in R^{n \times m}$ be a matrix containing all $n$ nodes with their features, where $m$ is the dimension of the feature vectors, and each $x_v \in R^m$ is the feature vector for $v$. $A$ and $D$ are the adjacency matrix and degree matrix of $G$, respectively. The formula for $D$ is as follows:

$$D_{ii} = \sum_j A_{ij} \tag{1}$$

Due to the self-loops, the diagonal element of $A$ is set to one1. One layer of GCN can only capture information about direct neighbors. When multiple GCN layers are stacked, information about larger neighborhoods will be integrated. One layer of GCN can be expressed as follows:

$$H^{(1)} = \sigma \left( \tilde{A} H^{(0)} W_0 \right) \tag{2}$$

$$\tilde{A} = D^{-\frac{1}{2}} A D^{-\frac{1}{2}} \tag{3}$$

where $\tilde{A}$ is the normalized symmetric adjacency matrix and $W_0 \in R^{m \times k}$ is a weight matrix. $\sigma$ is an activation function. $H^{(0)}$ is the matrix of hidden states in the l$^{\text{th}}$ layer and $H^{(0)} = X$. As mentioned previously, high-order neighborhood information can be merged by stacking multiple GCN layers. One layer of GCN can be expressed as follows:

$$H^{(j+1)} = \sigma \left( \tilde{A} H^{(j)} W_j \right) \tag{4}$$

where $j$ is the number of layers and $W_j$ is the layer-specific trainable weight matrix.

Multiple GCN layers can be stacked to capture high-order relations in the graph. We considered using three layers of GCN for semi-supervised node classification in this paper [34]. Our forward model takes the form:

$$V = \tilde{A} \, \text{ReLU} \left( \tilde{A} \, \text{ReLU} \left( \tilde{A} X W_0 \right) W_1 \right) W_2 \tag{5}$$

where $X$ is the input matrix with one-hot representation and $V$ is the representation matrix for all nodes in the graph.

## 4. Personality Graph Convolutional Networks

In this section, we introduce the personality GCN in detail. The framework of the model is depicted in Figure 1. It consists of three layers: (1) the embedding layer, where our personality GCN is initialized with one-hot representations of users, words, and documents and then learns the embedding of users, words, and documents together under the supervision of known user and document personality labels; (2) in the GCN layer, we use graph convolution to extract more deep interaction features from the constructed graph; here, we construct an information interaction graph to model the potential relationship structure between the three types of nodes; (3) the classification layer, where we feed the deeper common information features into five classifiers after the fully connected layer. In the following section, we describe the details of these layers.

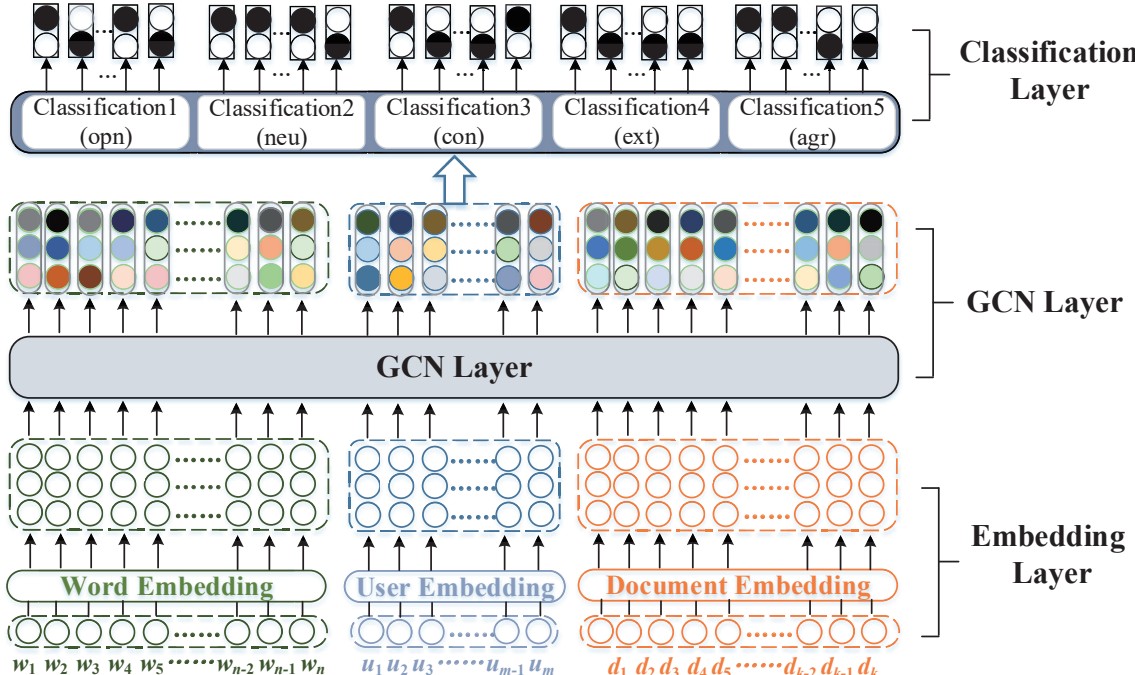

**Figure 1.** The architecture of personality GCN.

### 4.1. GCN Layer

Embedding layer: The initial input vector of the personality GCN is that the users, words, and documents are all represented by one-hot encoding, and we used the one-hot embedded form to embed all the information for all nodes.

Construct graph: We represent the relevant relationships as an information graph, as shown in Figure 2. The personality information graph $G = (V, E)$, which is composed of several different types of vertices and edges, is defined as follows:

- Let $U \in V$ denote a collection of textual users. Each user has their own personality labels. In the myPersonality dataset, each user completed a questionnaire and was assigned one or more personality traits based on the answers. In the essays dataset, these are authors with personality labels, which are self-assessments, derived from z-scores calculated by Mairesse et al. [26], converted from the scores to nominal categories by Celli et al. [22], and split by the median.

- Let $D \in V$ denote a collection of each piece of text information. The text was produced by users who participated in the Big Five personality assessment. Since it is two different datasets, the text includes two types of user state information and a user's stream of consciousness text information. Each text information represents a document.

- Let $W \in V$ denote the set of words in the corpus.

The graph vertices are connected via a set of edges described hierarchically as follows:

- $E_{UD} \in E$: All text authors have a connection with the text they have written. Note that text authors are only associated with their own written text.

- $E_{DW} \in E$: All text is related to the words contained in the text. That is, the relationship between different texts can be established by the same word appearing in the text.

- $E_{WW} \in E$: There is a certain correlation between all words, and we used pointwise mutual information (PMI) to compute the weight between two words. The weight between them is the edge between them.

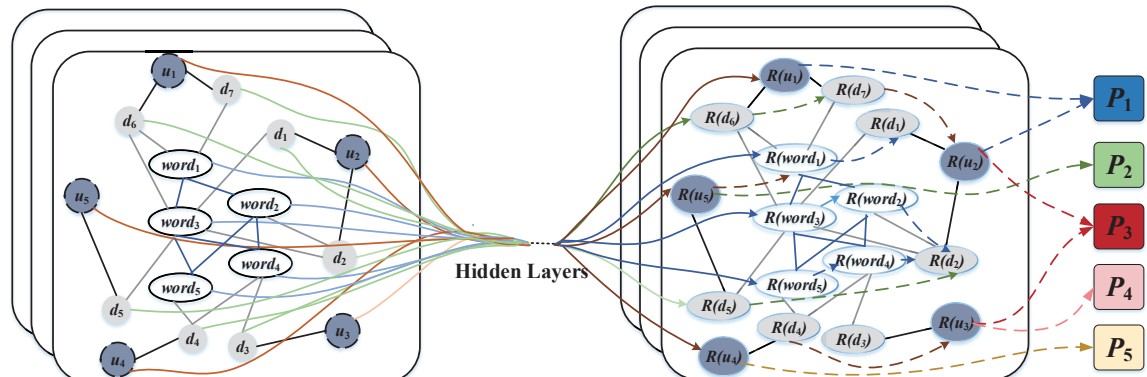

**Figure 2.** Constructed graph of personality GCN. Nodes beginning with "*u*" stand for user nodes, "*d*" for document nodes, and "*w*" for word nodes. Black edges indicate user-document edges, gray edges document-word edges, and blue edges word-word edges. $R(x)$ denotes the representation (embedding) of $x$. $P_1$: openness, $P_2$: neuroticism, $P_3$: conscientiousness, $P_4$: extroversion, $P_5$: agreeableness.

Global information transfer: This paper constructs a large and heterogeneous personality information graph, which contains user nodes, document nodes, and word nodes. It can not only explicitly model the co-occurrence of global words, but also easily adapt to graph convolution, as shown in Figure 2. We built edges between user nodes (user-document edges), documents (document-word edges), and words (word-word edges) in the entire corpus. The edge weight between the user node and the document node is whether the document is written by the user. The term frequency inverse document frequency (TF-IDF) of the word in the document is used to calculate the edge weights between document nodes and word nodes. The term frequency (TF) is the number of times a word appears in the document, and the inverse document frequency (IDF) is the number of documents containing the word log inverse scale. The paper [37] found that using TF-IDF weights was better than using only the term frequency (TF). To take advantage of global word co-occurrence information, we used a fixed-size sliding window on all documents in the corpus to collect co-occurrence statistics. We used pointwise mutual information (PMI) to calculate the weight between two word nodes. This is a popular measure of word association. Generally, the edge weight between node $i$ and node $j$ is defined as:

$$f(i,j) = \begin{cases} 1 & \{i,j : i = j\} \cup \{i,j : i \in U, j \in D, E_{UD} \in E\} \\ TF-IDF_{(ij)} & \{i,j : i \in D, j \in W, E_{DW} \in E\} \\ PMI(i,j) & \{i,j : i \in W, j \in W, PMI(i,j) > 0, E_{WW} \in E\} \\ 0 & \text{otherwise} \end{cases} \tag{6}$$

where $U$ is the set of users, $D$ is the set of documents, $W$ is the set of words. The PMI value is calculated as follows:

$$PMI(i,j) = \log \frac{p(i,j)}{p(i)p(j)} \tag{7}$$

$$p(i,j) = \frac{\#G(i,j)}{\#G} \tag{8}$$

$$p(i) = \frac{\#G(i)}{\#G} \tag{9}$$

where $\#G$ is the total number of windows in the corpus, $\#G(i)$ is the number of sliding windows in the corpus containing the word $i$, and $\#G(i,j)$ is the number of sliding windows in the corpus containing the words $i$ and $j$. A positive PMI value stands for the semantic relevance of the two words being high, while a negative PMI value stands for there being little or no semantic relevance of the two words. Therefore, we only added edges between two words with a positive PMI.

Figure 3 shows an example of how our personality GCN model aggregated and passed information from a node's local neighbors. The blue nodes represent users to do personality recognition. Green edges link to first-order neighbors, blue edges to second-order neighbors, and orange edges to third-order neighbors. The personality GCN can capture user, document, and word relationships and global word co-occurrence information. User and document nodes' label information can be passed through their neighbor nodes to other documents and words. Each node in the graph constantly changes its state until the final equilibrium due to the influence of neighbors and farther nodes. The closer the relationship is, the greater the influence of the neighbors.

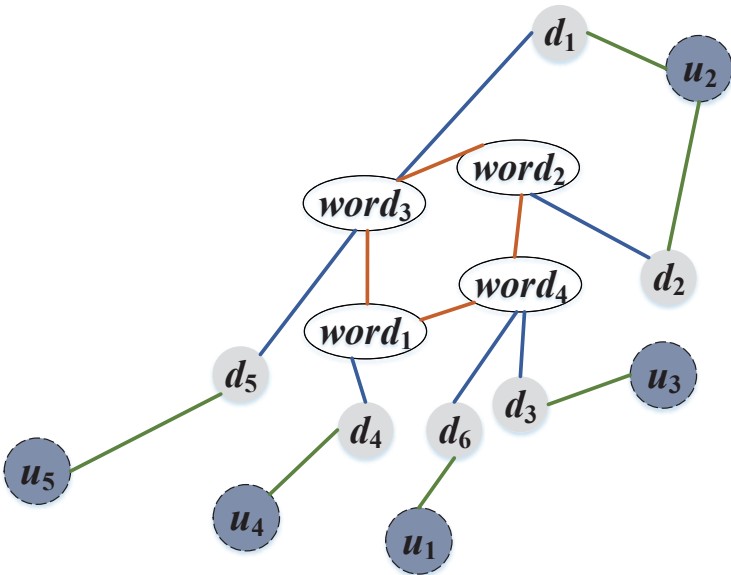

**Figure 3.** Example of unfolding of a GCN computational graph.

### 4.2. Classification Layer

The methods proposed by previous researchers are effective methods for detecting personality, but these methods use five independent classifiers to classify personality and do not consider the correlation of the different personality traits of a person. This is not accurate in personality recognition. Our model is based on the same set of information features at the time of final classification and backpropagates the classification results during information sharing to adjust the information features continuously to achieve the optimal state. Our model classifies user personality based on information sharing at the same time, rather than classification based on five types of features. This classification method fully considers the correlation between different personality traits of a person.

After constructing the personality information graph, we feed the information graph into a simple three layer personality GCN, as shown in Figure 2. The third layer nodes (user/word/document) embed the same dimension as the label set and feed five softmax classifiers:

$$Z = \text{softmax} \left( \tilde{A} \, \text{ReLU} \left( \tilde{A} \, \text{ReLU} \left( \tilde{A} X W_0 \right) W_1 \right) W_2 \right) \tag{10}$$

where $\text{softmax} \left( x_i \right) = \frac{1}{\mathcal{Z}} \exp \left( x_i \right)$ with $\mathcal{Z} = \sum_i \exp \left( x_i \right)$. The loss function is defined as the cross-entropy error over all labeled users:

$$\text{loss} = - \sum_{d \in Y_D} \sum_{f=1}^{F} Y_{df} \ln Z_{df} \tag{11}$$

where $Y$ is the label indicator matrix. $Y_D$ is the set of user indexes with labels, and $F$ is the size of the output features, which is the number of classes, where $F$ is five. In Equation (10), $E_1 = \tilde{A} X W_0$ contains the first layer of users, documents, and word embeddings, and $E_2 = \tilde{A} \, \text{ReLU} \left( \tilde{A} X W_0 \right) W_1$ contains the

second layer of users, documents, and word embeddings. Figure 2 shows the overall personality GCN model by means of information dissemination.

A three layer GCN allows messages to be passed between nodes up to three steps apart. Therefore, although there are no direct document-document edges and user-user edges in the graph, a three-layer GCN allows information to be exchanged between pairs of documents and users. Our experimental results showed that the three layer and four layer GCN performed better than the one layer or two layer GCN, and more layers did not improve the performance. This was similar to the result in [35].

## 5. Experiment

In this section, we evaluate our personality GCN on two public and authoritative datasets. The detailed information of the experimental equipment and resources in this paper is as follows: Intel e5-2650v4, 1T STAT, 32G, GPU 1080ti, 11G.

### 5.1. Datasets and Evaluation

(1) Datasets:

The experimental data in this paper were two different datasets: myPersonality [38] and essays. The first dataset, myPersonality, included data from 250 Facebook users who had approximately 9917 states and contained a given personality label based on the Big Five personality model. It included users' status information and external information (like release time, network size, etc.). In this paper, we only used the textual information of the users in the dataset. The personality trait distributions of the myPersonality dataset are shown in Table 2.

**Table 2.** Distributions of the myPersonality dataset.

| Value | OPN | CON | EXT | AGR | NEU |
|-------|-----|-----|-----|-----|-----|
| Yes   | 176 | 130 | 96  | 134 | 99  |
| No    | 74  | 120 | 154 | 116 | 151 |

The second dataset, essays, included 2468 anonymous essays that were labeled with personality traits for each author. We removed essays containing only the information "Err: 508" from the dataset and experimented with the remaining 2467 essays [39]. The personality trait distributions of the essays dataset are shown in Table 3.

**Table 3.** Distributions of the essays dataset.

| Value | OPN  | CON  | EXT  | AGR  | NEU  |
|-------|------|------|------|------|------|
| Yes   | 1271 | 1253 | 1276 | 1310 | 1233 |
| No    | 1196 | 1214 | 1191 | 1157 | 1234 |

All data went through a preprocessing stage before the experiment. The preprocessing steps included removing URLs, symbols, names, spaces, lowercase letters, stemming, stop words, etc. The data in Bahasa underwent an additional pre-processing process; this was replacement of slang words or non-standard words, which was manually executed. After finishing, they were then translated into English. In the experiment, both the validation and test sets were ten percent of the dataset. The data ratio of the training set, the validation set, and the test set was 8:1:1.

We removed words that appeared fewer than five times in the corpus. We found that deleting words that occurred less frequently helped improve performance and saved time. However, when deleting words that occurred fewer than ten times in the corpus, the performance of the model decreased. We think this was because words occurring nearly ten times belonged to a higher frequency, which could affect the performance of personality recognition.

(2) Evaluation:

Following the previous study [16,17,26,39–48], which achieved the best performance, we chose the accuracy and F1-score as the primary metrics for evaluating the performance of our models. Specifically, this paper reports the highest accuracy and F1-score of each trait for each dataset with each method. The accuracy rate is our most common evaluation index; generally, the higher the accuracy is, the better the classifier. Precision and recall indicators sometimes conflict, so they need to be considered comprehensively. The most common method is the F-measure (also known as the F-score). The F-score is the harmonic average of precision and recall. The calculation formulas of the F-score and accuracy are as follows:

$$\text{accuracy} = (TP + TN)/(TP + TN + FP + FN) \tag{12}$$

$$\text{precision} = TP/(TP + FP) \tag{13}$$

$$\text{recall} = TP/(TP + FN) \tag{14}$$

$$F - \text{score} = \frac{(\alpha^2 + 1) \times preci\text{sion} \times \text{recall}}{\alpha^2(\text{precision} + \text{recall})} \tag{15}$$

where $TP$ indicates the number of positive samples that are correctly predicted. $TN$ is the number of negative samples that are correctly predicted. $FP$ represents the number of negative samples that are wrongly predicted. $FN$ is the number of positive samples that are wrongly predicted. The most common $F1$ is $\alpha = 1$. Therefore, $F1$ combines the results of precision and recall. When $F1$ is higher, it can indicate that the test method is more effective.

$$F1 - \text{score} = \frac{2 \times preci\text{sion} \times \text{recall}}{\text{precision} + \text{recall}} \tag{16}$$

*5.2. Baselines Models*

We compared our personality GCN with multiple state-of-the-art personality recognition methods as follows:

(1) myPersonality dataset baseline:

- Machine learning: This article [16] explores the use of machine learning techniques to infer the personality characteristics of users from their Facebook status updates.
- Pearson's r: Personality features were selected based on the Pearson correlation coefficient analysis method [17].
- Particle swarm optimization (PSO): This paper [17] introduced a feature selection technique called PSO, which can find the best subset of features.
- Linguistic features: This method [40] uses language functions such as LIWC [41], SPLICE [42], and the SNAfunction [43] to identify user personality.
- GloVe word embeddings: This method [40] automatically explores the dataset to find the relationship between words and personality. In this study, GloVe was used for word embeddings [44].
- Feature combination: The purpose of this article was to study the predictability of personality traits of Facebook users by combining the different types of features and measures of the Big Five model [45].
- SMOTETomek: A personality recognition method that combines the synthetic minority oversampling technique with the Tomek link resampling technique [46].

(2) Essay dataset baseline:

- Word n-grams: As the baseline feature set, the work in [47] used 30,000 features: the 10,000 most frequent word unigrams, bigrams, and trigrams in the essay dataset.
- Mairesse: Francois Mairesse and colleagues used their method [26], in addition to LIWC and other features, like imageability, to improve performance, which they called the Mairesse baseline.
- Machine learning: In this paper [16], they explored the use of machine learning techniques to infer the personality traits of authors from their essays.
- Information gain (IG): The information gain of all 80 features is calculated for each Big Five dataset. For each dataset, only features with nonzero information gain are selected [48].
- Principal component analysis (PCA): After implementing PCA on all 80 LIWC features of Weka, a total of 56 feature vectors were found and used to create new datasets for each of the Big Five datasets [48].
- CNN + Mairesse: Feature combination of features extracted from CNN and Mairesse features [39].
- PSO: This paper [46] proposed a PSO-based personality recognition method that can optimize features for each group of features.

*5.3. Experimental Results Analysis*

(1) Performance comparison with existing methods:

The results of the myPersonality and essay dataset classification are summarized in Tables 4–7. We report the accuracy and F1-score of personality recognition. Note that we report only the best results from existing models [16,17,26,39–48]. The results in our model used the parameters that achieved the best results in the dataset. The results clearly showed that personality GCN outperformed the other methods in supervised personality recognition settings.

Tables 4 and 5 provide the accuracy and F1-score in the myPersonality dataset comparison between the existing model and our methods. From Table 4, we can see that the personality GCN model was 0.49% lower in the personality trait of neuroticism (NEU) and was higher than the existing model in the remaining four personality traits and the overall average. As seen in Table 5, although our model was 0.07 and 0.04 lower in agreeableness (AGR) and openness (OPN) traits, respectively, our average level was higher than the existing model of 3.6%. The highest accuracy and F1-score obtained from personality GCN were 80% and 0.85, respectively. The highest average accuracy and F1-score were 76.6% and 0.768, respectively, also from personality GCN.

**Table 4.** Personality GCN classification results of accuracy on the myPersonality dataset.

| Model | Traits | | | | | |
|---|---|---|---|---|---|---|
| | EXT | NEU | AGR | CON | OPN | Average |
| Linguistic features | 68.80% | 60.80% | 63.20% | 59.20% | 70.40% | 64.48% |
| GloVe word embeddings | 78.90% | 79.49% | 67.39% | 62.00% | 79.31% | 73.43% |
| Feature combination | 78.60% | 68.00% | 65.30% | 69.80% | 73.30% | 71.00% |
| SMOTETomek | 76.00% | 73.50% | 74.50% | 65.00% | 76.00% | 73.00% |
| Personality GCN | **80.00%** | 79.00% | **68.00%** | **76.00%** | **80.00%** | **76.60%** |

The accuracy and F1-score in the essay dataset comparison between the existing model and our methods are shown in Tables 6 and 7. As can be seen from Tables 6 and 7, except for the F1-score of OPN, our model performed better than the previous model in both accuracy and F1-score. The highest accuracy and F1-score obtained from personality GCN were 64.8% and 0.69, respectively. The highest average accuracy and F1-score were 60.92% and 0.68, respectively. In other words, the experimental results showed that the personality GCN method achieved the highest average accuracy and F1-score

in the two datasets and could effectively improve the accuracy and F1-score compared with other methods [16,17,26,39–48].

**Table 5.** Personality GCN classification results of F1-score on the myPersonality dataset.

| Model | Traits | | | | | |
|---|---|---|---|---|---|---|
| | EXT | NEU | AGR | CON | OPN | Average |
| Machine learning | 0.65 | 0.53 | 0.50 | 0.55 | 0.62 | 0.57 |
| Pearson's r | 0.69 | 0.61 | 0.63 | 0.64 | 0.66 | 0.646 |
| PSO | 0.77 | 0.71 | 0.77 | 0.69 | 0.79 | 0.746 |
| SMOTETomek | 0.73 | 0.73 | 0.75 | 0.63 | 0.82 | 0.732 |
| Personality GCN | **0.85** | **0.79** | 0.70 | **0.75** | 0.78 | **0.768** |

As can be seen from Tables 4–7, the three personality traits OPN, extroversion (EXT), and NEU were very closely related to the text information, and the features of OPN, EXT, and NEU could be well extracted from the text information. From another perspective, the feature information of personality traits AGR and conscientiousness (CON) in text information was not obvious. The experimental results of the two datasets showed that the personality GCN showed better performance in the short text dataset. We guessed that short text could be better learned than long text, and for long text, it was easy to lose information during the learning process.

**Table 6.** Personality GCN classification results of accuracy on the essay dataset.

| Model | Traits | | | | | |
|---|---|---|---|---|---|---|
| | EXT | NEU | AGR | CON | OPN | Average |
| Word n-grams | 51.72% | 50.26% | 53.10% | 50.79% | 51.52% | 51.48% |
| Mairesse | 55.13% | 58.09% | 55.35% | 55.28% | 59.57% | 56.68% |
| IG | 54.74% | 57.54% | 57.54% | 55.55% | 61.83% | 57.44% |
| PCA | 55.75% | 58.31% | 56.71% | 57.30% | 59.38% | 57.49% |
| CNN + Mairesse | 58.09% | 59.38% | 56.71% | 57.30% | 59.38% | 58.17% |
| Personality GCN | **60.00%** | 63.00% | **57.70%** | **59.10%** | **64.80%** | **60.92%** |

**Table 7.** Personality GCN classification results of F1-score on the essay dataset.

| Model | Traits | | | | | |
|---|---|---|---|---|---|---|
| | EXT | NEU | AGR | CON | OPN | Average |
| Machine learning | 0.600 | 0.570 | 0.580 | 0.560 | 0.630 | 0.588 |
| IG | 0.541 | 0.574 | 0.557 | 0.55 | 0.618 | 0.568 |
| PCA | 0.556 | 0.583 | 0.563 | 0.560 | 0.619 | 0.576 |
| PSO | 0.620 | 0.630 | 0.620 | 0.650 | 0.700 | 0.644 |
| Personality GCN | **0.670** | **0.690** | **0.690** | **0.680** | 0.670 | **0.680** |

(2) Performance with different personality GCN layers:

Figures 4 and 5 show that the number of personality GCN layers played an important role in the process of information propagation. In the one-layer personality GCN, the information only flowed between neighbors, so for our user-document-word tripartite graph, the number of personality GCN layers was at least three, so that the information could flow from the user to the word node. To better understand this, we kept the parameters in the experiment consistent and drew Figures 4 and 5 to display the relation between the personality GCN layer and the performance (accuracy and F1-score) of our model.

We report the accuracy and F1-score in two datasets under different GCN layers $L$ among [1,2,3,4,5]. It is worth noting that we only drew the line when $L \leq 5$ because there was no improvement when $L > 5$. From Figures 4 and 5, we can see that the one-layer GCN had a relatively poor performance on both datasets, while when the number of layers was three or four, both datasets achieved the best performance, which illustrated that the one-layer GCN could not transfer information from the user node to the word node and that at least three layers of the GCN could capture the word co-occurrence between the user node and the word node. When $L > 4$, the accuracy and F1-score declined to different degrees between the two datasets, and we assumed that the word representation was distorted due to too many instances of information propagation.

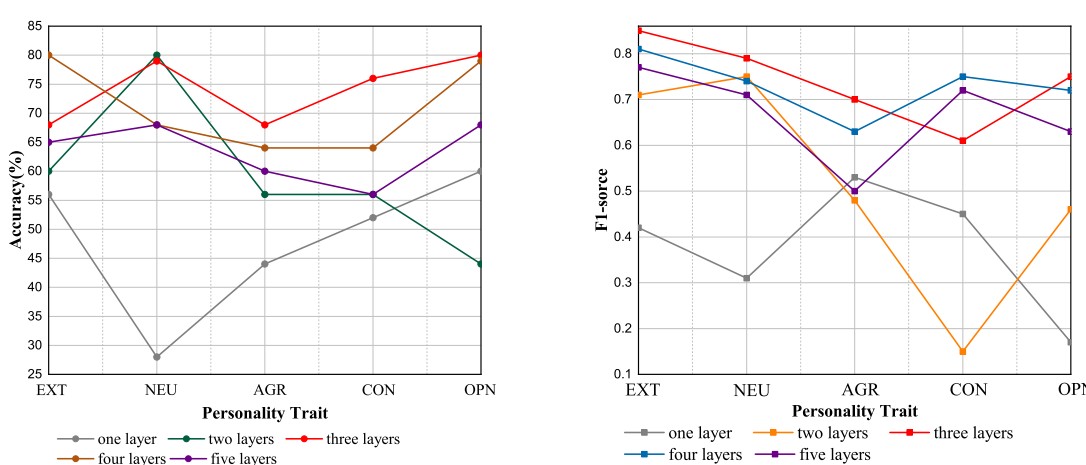

**Figure 4.** Accuracy and F1-score in the myPersonality dataset with different numbers of personality GCN layers.

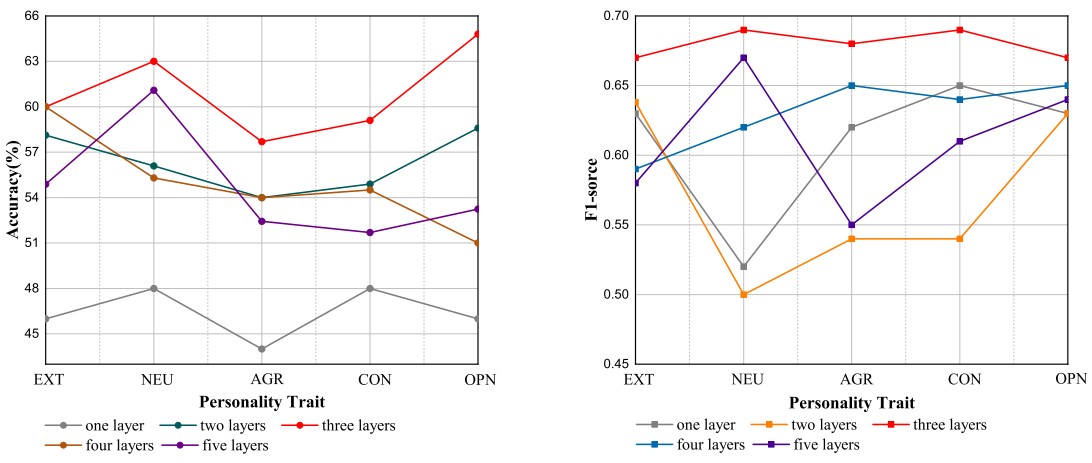

**Figure 5.** Accuracy and F1-score in the essay dataset with different numbers of personality GCN layers.

(3) Discussion of personality GCN:

From the experimental results, it could be seen that the proposed personality GCN could improve personality recognition results. There were two main reasons why personality GCN performed well. (1) The personality graph could capture user-document relationships, document-word relationships, and word-word relationships and use information transfer and word co-occurrence to achieve global information sharing. (2) When identifying user personality, we considered that there was a certain correlation between the five personality traits of a person. Instead of using five separate sets of

feature information, we shared a set of information functions to classify the five personality traits. Considering the correlation between the five personality traits also helped to improve the accuracy of personality recognition.

## 6. Conclusions

In this paper, we proposed a novel personality recognition method called personality GCN. We constructed a heterogeneous graph of user document words for the entire corpus. Personality GCN could capture global word co-occurrence information and make good use of limited labeled users. We used shared features to incorporate the correlations between the five personality traits into the model. A simple three-layer or four-layer personality GCN demonstrated promising results by outperforming numerous state-of-the-art methods on two public and authoritative datasets.

In addition to the GCN model used in this paper, we hope to try different word embeddings methods to embed nodes. We plan to combine BERT's word embeddings with GCN to identify personality.

At the same time, from another perspective, protecting personal privacy is also an important issue to be considered. We will provide some examples when people try to conceal their personality in your future tests; for example, hiding personality information in data, replacing feature words related to personality traits, confusing source data related to personality information, and so on.

**Author Contributions:** Conceptualization, Z.W. and C.-H.W.; data curation, B.Y.; methodology, Z.W. and Q.-B.L.; validation, Q.-B.L. and B.Y.; writing, original draft, Z.W.; writing, review and editing, Z.W.; project administration, C.-H.W and K.-F.Z.; funding acquisition, K.-F.Z. All authors read and agreed to the published version of the manuscript.

**Funding:** This research was funded by the National Key R & D Program of China Grant Numbers 2017YFB0802703.

**Conflicts of Interest:** The authors declare no conflict of interest.

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
