# Peer review of "Encoding Text Information with Graph Convolutional Networks for Personality Recognition"

_applsci, doi:10.3390/app10124081_

Round 1
Reviewer 1 Report
General Overview:
This work talks about the implementation of GCN for personality recognition using personality traits data related to the Big Five model and text from different users retrieved from two public datasets. The important keys of this works are the implementation of the GCN taking into account the correlation between personalities to help improving the performance of the recognition task.
This is a very interesting and developing field in personality recognition. This work is worthy of publication; however, a few points need to be considered first.
- Some grammar/spelling/structural issues throughout manuscript. Some examples are:
Line 20 whose thought patterns? – of a person’s/user’s thought patterns.
Line 22 etc.,
Line 58, to our understanding.
Line 81 indicators than indicate.
- More concise details about the data preprocessing, the tools’ specification to implement the models, and discussion about the nature of the data used related to the results obtained.
- Both datasets used are not related to social information, but they are related with text and word corpus, I believe that the title
Abstract:
It lacks on mention the context of the personality profile for the recognition task: work, online presence, daily routine?
Personality based on? Which personality theory.
line 9 users of what exactly?
Which 5 personality traits, there is different approaches under the psychological perspective, it will be necessary to clarify the personality model used for this work in the abstract.
Introduction:
I think that before statement in line 21, it is important to define personality detection or recognition because, next sentences are related to examples in where personality recognition can be applied to enhance certain task (and no personality as the paragraph imply).
Line 31 ‘most popular’ should be omitted.
Line 32 Big five personality traits model.
Line 34, there are still some concerns about how this model does not fit the whole personality spectrum and there are other proposed theories that counter this particular model. What does the actors mean when they say well-experimented and well-scrutinized?
Line 35 If personality is a human characteristic that exhibit similar behavior patterns across time, how can it vary from each situation? It is not more suitable to say that the certain characteristics or personality come up or manifest depending of the situation?
Line 37 Text data is widely use in the personality recognition field, but here the authors do not explain why is important, what kind of characteristics and behavioral patterns the text exhibit? I suggest to the authors concisely explain this statement.
Line 54 social information from?
Line 59 what does social information corpus means in the context of this work? It is still not clear exactly form where or which information is used to perform the personality recognition until this point in the document.
Related work:
Line 80 personality in singular, not plural if we are talking about a single individual.
Background:
It will be useful for the reader to describe the datasets used and to comment the nature of the data, instead of being part of the experiment section, or allocated before the explanation of the GCN model construction.
Personality graph convolutional networks:
Line 201 different personality traits, not different personalities.
In the document nodes besides distinguish between the number of documents belonging to each subject in one dataset or the other (social presence on Facebook or essays), they have another kind of characteristics or labels like for example topics of each of the documents?
Experiment:
There is not reference related to the dataset “essays”, and also there is not mention about the permission to use the data in the document.
Because of the nature of the datasets, the pre-processing step is not that clear, only words related to subjects, adverbs, verbs, objects were included? What about conjunctions and what about the language of each text? It will be worth of clarification some specific details that help to understand how the nodes were built specifically for the word nodes.
In line 222 I believe that the number of observations is 205 and no 250 according to table 2. Also because not all the observations have all 5 personality values as tables 2 and 3 shows, how does the correlation for this cases work in the recognition process?
It will be important to include the technical information about the equipment and resources used to perform the experiment.
In tables 4 and 5, and 6 and 7 there is a particular reason why in one the accuracies are shown as percentage and in the other are shown as decimals if both are referring as accuracy scores? I believe that the table’s titles correspond one to accuracies and the other to F1-scores.
Discussion and conclusion:
The discussion is too short, the authors do not talk about why for some traits, the accuracy and F1-scores were higher than the ones reported in other works (for example for neuroticism, openness and agreeableness in each context). Also it is not mention how the nature of the data could affect the personality recognition process.
It will be worth to discuss about how the techniques used to calculate the weights on the edges can affect the performance of the GCN in contrast with other works.
It will be beneficial if the authors also discuss the implications of using this model in other corpus with different languages or context were amount of words and grammatical structure differs from English.
Reviewer 2 Report
Dear authors!
Your interdisciplinary study is quite interesting. Some corrections are needed, however.
Quote: "We propose a novel personality recognition method based on the graph convolutional neural network method. According to my understanding.."
First, the novelty of this approach should be explicitly clarified. I found many similar studies, including book chapters (https://link.springer.com/chapter/10.1007/978-3-319-77116-8_23) and more. The idea to apply convolutional NN to personality recognition is not new.
Second, it is strange to use "my" when there are many authors listed.
Quote: "This paper focuses on the correlation between the five personality traits."
Please, explain why this 5-trait model was chosen as the core of your study. There are a lot of psychological schools and personality models. This choice is crucial and therefore should be better grounded.
The first author's ORCID link is void and email is e-mail@e-mail.com.
The accuracy results are not impressive, especially in Fig. 5.
Conclusions section is too short, weak, and barely supported by results.
Round 2
Reviewer 2 Report
Thank you for revising your paper. Despite that I cannot observe the changes in the manuscript because they are not marked or highlighted, I am satisfied with the answers and can recommend the paper for publication after some minor edits including English style and grammar. Also please expand the conclusions section. I recommend to include some examples when people try to conceal their personality in your future tests.
